

# Synthesis of geological data and comparative phylogeography of lowland tetrapods suggests recent dispersal through lowland portals crossing the Eastern Andean Cordillera

Erika Rodriguez-Muñoz[1], Camilo Montes[2], Fernando J. M. Rojas-Runjaic[3,4] and Andrew J. Crawford[1]

[1] Department of Biological Sciences, Universidad de Los Andes, Bogotá, DC, Colombia
[2] Department of Physics and Geosciences, Universidad del Norte, Barranquilla, Atlantico, Colombia
[3] Fundación La Salle de Ciencias Naturales, Museo de Historia Natural La Salle, Caracas, Venezuela
[4] Laboratório de Herpetologia, Coordenação de Zoologia, Museu Paraense Emílio Goeldi, Belém, Pará, Brazil

## ABSTRACT

Vicariance is the simplest explanation for divergence between sister lineages separated by a potential barrier, and the northern Andes would seem to provide an ideal example of a vicariant driver of divergence. We evaluated the potential role of the uplift of the Eastern Cordillera (EC) of the Colombian Andes and the Mérida Andes (MA) of Venezuela as drivers of vicariance between lowland populations co-distributed on both flanks. We synthesized published geological data and provided a new reconstruction showing that the EC-MA grew from north to south, reaching significant heights and separating drainages and changing sediment composition by 38–33 million years ago (Ma). A few lowland passes across the EC-MA may have reached their current heights (~1,900 m a.s.l.) at 3–5 Ma. We created a comparative phylogeographic data set for 37 lineages of lowland tetrapods. Based on molecular phylogenetic analyses, most divergences between sister populations or species across the EC-MA occurred during Pliocene and the Quaternary and a few during the latest Miocene, and coalescent simulations rejected synchronous divergence for most groups. Divergence times were on average slightly but significantly more recent in homeotherms relative to poikilotherms. Because divergence ages are mostly too recent relative to the geological history and too asynchronous relative to each other, divergence across the northern Andes may be better explained by organism-environment interactions concomitant with climate oscillations during the Pleistocene, and/or dispersal across portals through the Andes.

Corresponding author
Erika Rodriguez-Muñoz,
e.rodriguez1548@uniandes.edu.co

# INTRODUCTION

Vicariance is a biogeographical model describing the division of a widespread ancestral population into two daughter populations through the appearance of a barrier to gene flow that eventually leads to allopatric speciation (*Crisci, 2001*). Such barriers are often hypothesized to arise through geological changes, such as mountain uplift or the appearance of rivers. Environmental changes, such as aridification, can also promote spatial isolation of a formerly continuous population or meta-population system (*Lomolino, Riddle & Whittaker, 2017*). Most older methods of biogeographic analysis are based on a strictly vicariance model because dispersal models were regarded as difficult to falsify as they are idiosyncratic and do not generate predictions for co-distributed species (*Rosen, 1978*; *Morrone & Crisci, 1995*). Vicariance models, on the other hand, suggest area cladograms that are testable with co-distributed species (*Ronquist, 1997*). Vicariance is also the most general and easiest to satisfy model of speciation, requiring only physical separation and the accumulation of nucleotide substitutions over time to create two daughter species from a common ancestor (*Mayr, 1963*; *Wiens, 2004*).

Despite the potential ubiquity of vicariance in promoting diversification of evolutionary lineages, many appreciable barriers, such as tall mountain ranges, appear to have conspecific populations on both sides. The effectiveness of a potential montane barrier may depend on the steepness of the environmental gradient and not on elevation *per se* (*Janzen, 1967*), suggesting that environmental heterogeneity might limit dispersal just as much, or perhaps more so, than simple physical obstruction. Therefore, the effectiveness of mountains in separating populations may have more to do with the interaction between environmental heterogeneity and organismal life history (*Paz et al., 2015*). For example, homeothermic animals likely have a higher tolerance to temperature heterogeneity compared to poikilotherms (*Porter & Gates, 1969*; *Ghalambor et al., 2006*). Life history variables could thus explain why the same potential barrier could differentially impact even closely related species. We evaluate the vicariant model by looking at 37 lineages of lowland tetrapods (amphibians, non-avian reptiles, birds, and mammals) distributed on either side of the Eastern Cordillera of the northern Andean mountains of South America, specifically Colombia and Venezuela. Tropical mountains are expected to represent an even stronger barrier to the local fauna and flora than temperate mountains because seasonal variation in temperature is relatively low (*Janzen, 1967*; *Ghalambor et al., 2006*). In addition, because the temperature in the tropical lowlands is relatively homogeneous spatially, species are expected to evolve narrower temperature tolerances compared to species that occupy high elevation zones and experience higher daily variations in temperature (*McCain, 2009*).

In this contribution we evaluate the potential role of the uplift of the Eastern Cordillera of the Colombian Andes (EC) and the Cordillera de Mérida in the Andes of Venezuela (aka, Mérida Andes, MA) as drivers of vicariance between lowland populations co-distributed on both flanks. We synthesize published geological data on the timing of uplift for the EC-MA and combine new mitochondrial DNA data with published sequences to create a data set of 37 lineages of lowland tetrapods. We focus on divergences
within taxonomic species, between sister species, and other relatively recent divergences. Even if mountain uplift does drive vicariant speciation, ancient lineages on either side of the Andes may not have been impacted by orogeny simply by virtue of being older than the mountains. Thus, the strongest test of the potential role of the EC-MA as a driver of vicariant divergence or speciation among lowland taxa is provided by the more recent divergences. Because we are interested in the potential effect of mountain uplift on lowland taxa, we avoid sampling high-elevation taxa whose diversification is likely driven by distinct processes (*Cadena et al., 2012*). We reconstructed time-calibrated phylogenies for each lineage to estimate age of gene divergence across the Andes. We performed a comparative phylogeographic analysis using hierarchical approximate Bayesian computation (hABC) to test for synchronous vicariance across the EC-MA and to estimate population divergence times. To evaluate the potential role of life history in explaining any variation in divergence times among lineages, we evaluated 13 general linear models (GLM) containing up to six variables.

## Surface uplift and speciation

Andean uplift triggered changes in climatic and hydrologic conditions across much of the continent, and this uplift is widely cited as having caused the primary divergence between ancestral populations on either side of the Andes. The rise of the EC, for example, is thought to have mediated the initial split between eastern and western populations of woodcreepers (*Dendrocincla*; *Weir & Price, 2011*), and between populations of the cane toad, *Rhinella marina*, with its recently resurrected sister species, *R. horribilis* (*Slade & Moritz, 1998*). However, these and similar phylogeographic studies make strong assumptions about the timing and location of uplift of the EC and the MA, which we refer to here together as the EC-MA (Fig. 1). *Hoorn et al. (2010)* shows that Andean uplift was a gradual process that occurred during more than 20 million years; however, many biologists are unaware of or misinterpret current evidence and assume that the Andes, or even a small section of the Andes such as the EC-MA, rose up synchronously across its entire length, without allowing for spatial heterogeneity in the timing of uplift. Biologists comparing molecular phylogenetic dating of a given clade to historical models of orogeny also need to make assumptions about when during the process of uplift did migration end. In the case of the EC-MA one could assume the age of initial uplift of roughly 20 million years ago (Ma) for the separation of an ancestral population, or perhaps the final uplift age of 5–2 Ma based on palynological records near Bogotá in the middle of the EC (*van der Hammen, Werner & van Dommelen, 1973*; *Hooghiemstra, Wijninga & Cleef, 2006*), or an intermediate age (for a revised palynological interpretation see *Molnar & Pérez-Angel (2021)*). With such a long time period between the start and end of uplift of the EC-MA, this process could explain nearly any divergence date investigators might obtain from evolutionary genetic analysis of their taxon of interest. Furthermore, some conspecific populations separated by the EC are not reciprocally monophyletic, demonstrating either incomplete lineage sorting caused by a young separation age or continued migration over the mountains (*e.g.*, *Miller et al., 2008*). Third, even if uplift were synchronous, and even if the researcher could connect without error a certain uplift age to divergence ages based on

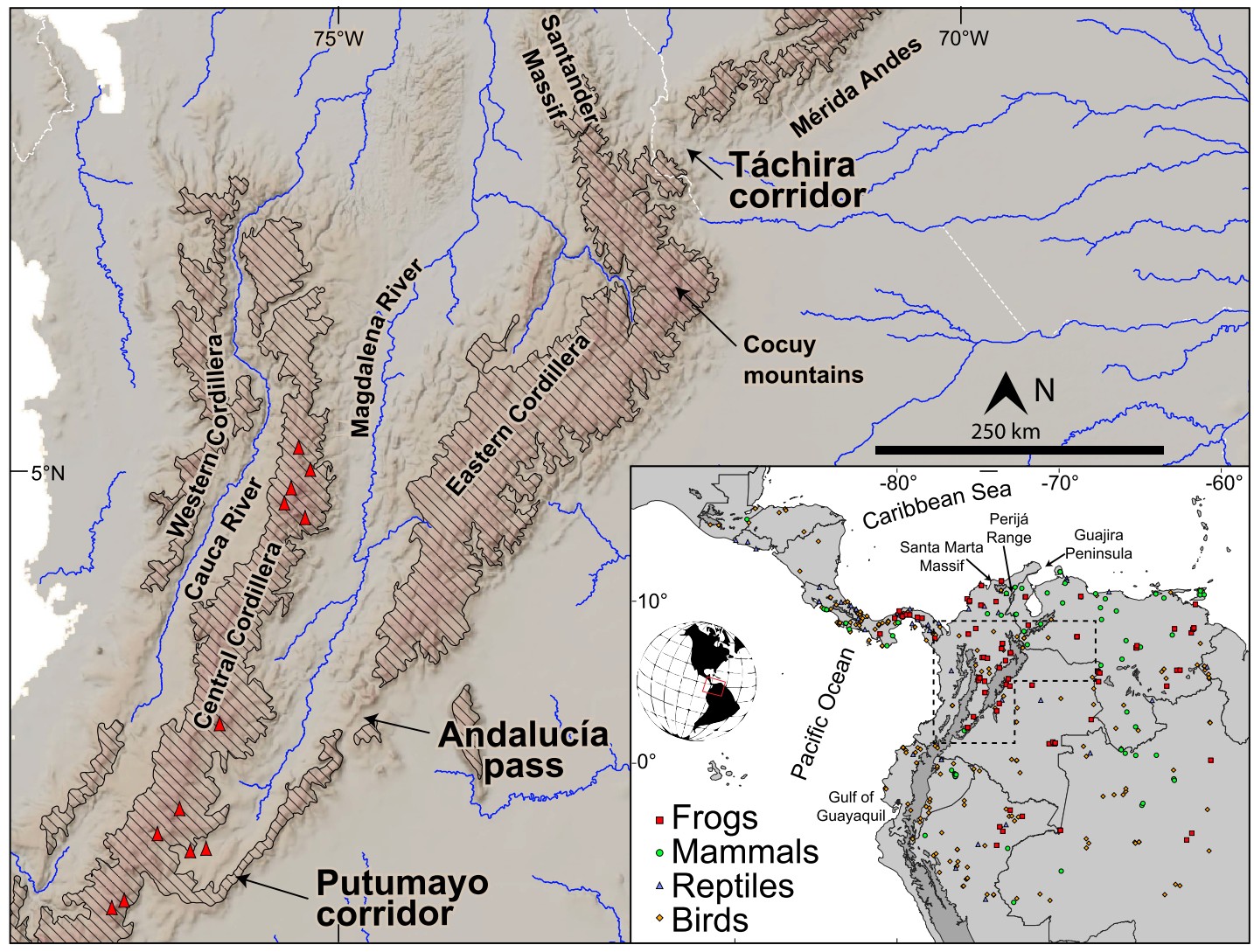

**Figure 1 Topography and main drainages in the northern Andes. Red triangles represent Pliocene and younger volcanic edifices.** Diagonal hatching represents elevations greater than 2,000 m. Inset shows sample localities from across South America and Central America per each of the four taxonomic classes of tetrapod studied here. The Eastern Cordillera (EC) and its northern extensions, the Santander Massif and Perijá Range (see inset), extend ~1,200 km from southern Colombia to northwestern Venezuela with summits reaching 5,300 m a.s.l. The Mérida Andes (MA) extend ~350 km to the NE reaching maximum elevations of ~5,000 m a.s.l. The EC joins the Central Cordillera across a low-elevation bridge (~2,000 m a.s.l., between the Putumayo corridor and Andalucia pass), while it splits from the MA at the low-elevation Táchira corridor (~1,000 m a.s.l.), constituting the lowest points of the eastern Andes.

genetic data, this age of separation between eastern and western populations might not be relevant to other organisms with contrasting life histories. Thus, any evaluation of the role of the Andes in driving primary divergence, *i.e.*, vicariance, requires recognizing explicitly that even a section of the Andes, such as the EC-MA, is composed of historically independent geological elements (*Hoorn et al., 2010*; *Boschman, 2021*), and it requires a widely comparative approach to capture the variation among lineages in their response to changes in environmental conditions.

To evaluate the role of the uplift of the EC-MA mountain chains on genetic variation, we chose to work with lowland terrestrial vertebrates as these share the synapomorphies that unite tetrapods, yet possess a variety of life history characteristics (size, type of locomotion, thermal physiology), and ample phylogeographic data sets are available online. By including phylogeographic data from mammals, birds, non-avian reptiles, and amphibians, we are able to evaluate the possible effect of variation in morphological and eco-physiological traits on divergence times across the EC-MA. We used comparative phylogeography and hierarchical approximate Bayesian computation (hACB) to test the following hypotheses, presented here in order from naively simple expectations to models of increasing complexity. Thus, (H0) is effectively our 'null' hypothesis that the separation of clades within each species or genus located east and west of the EC-MA was mediated by northern Andes uplift, reflecting simultaneous divergences among all taxa studied, supporting a strict (classic) vicariance model and little role of life-history variation affecting the timing of divergence. This null hypothesis predicts that (A) our hABC test for synchronous vicariance will support a single temporal interval containing all divergence events. According to oft-cited estimates, the divergence time interval would be around (B) 5–2 Ma (but see our synthesis of geological evidence below). Only if our study has sufficient power to reject H0 do we continue with the next three hypotheses. These suggest that barriers are the result of organismal-environmental interactions, implying that the effect of the rise of the EC-MA on each species depends on its eco-physiological traits. (H1) The separation of eastern and western clades was asynchronous among taxa and this variation can be explained by differences in elevation and geographic range. As the EC-MA rose up, lowland taxa would have separated first. Furthermore, organisms with wider distributional ranges, higher dispersal abilities, and/or broader tolerance of landscape heterogeneity should have been less affected by the environmental gradient generated by mountain uplift. H1 thus predicts that species with higher elevation ranges and widespread species will show younger divergence times relative to lowland and narrowly distributed species since the latter may have lower dispersal abilities or more specific habitat requirements. (H2) The effectiveness of a montane barrier will be stronger for small-bodied species since they are poor dispersers compared to bigger species (*Jenkins et al., 2007*; *Paz et al., 2015*), predicting that divergences vary according to body size, with older divergences in smaller species. (H3) Variation in divergence times (if any) can be explained by physiological traits, as follows. (3A) homeotherms will show younger divergences across the Andes, while poikilotherms will show older divergences due to their susceptibility to environmental temperature variation, reducing their dispersal across the incipient environmental gradient created by mountain uplift (*Porter & Gates, 1969*; *Buckley, Hurlbert & Jetz, 2012*). (3B) Flying species (bats and birds) will show younger divergences with respect to non-volant species since the former should disperse much more readily over the nascent Andes. In summary, H1 involves species' ranges, H2 focuses on body size, and H3 on tetrapod physiology and locomotion.

## Geology of the Andes

Here we consider the northern Andes as the set of mobile blocks west of the Guiana Shield north of the Gulf of Guayaquil (*Pennington, 1981*; *Pindell & Kennan, 2009*). This set of blocks includes three mountain chains (Fig. 1): The Western Cordillera (WC), the Central Cordillera (CC), and the Eastern Cordillera (EC). To approach the history of elevation of these mountain ranges we need to establish a base level from which to study the influence of elevation, the development of terrestrial barriers, and the effects of a dynamic landscape on biotas and diversification. A convenient base level is the Upper Cretaceous sequence deposited ~100 to 66 Ma, a time when most of northern South America was stable and covered by shallow seas (*Villamil & Pindell, 1998*; *Moreno-Sanchez & Pardo-Trujillo, 2003*; *Boschman, 2021*). The record of these shallow marine conditions consists of limestone and fine-grained sedimentary rocks such as black shale (*Villamil & Arango, 1998*). Along the coastline of this seaway, westward-flowing river systems predating the Amazonian/Orinoco drainages, built large sandy deltas today exposed, for instance in the Cocuy mountains of Colombia (Fig. 1, *Fabre, 1985*). These Upper Cretaceous sequences are preserved in the Mérida Andes (MA), the Guajira peninsula, the Santa Marta massif, the Perijá range, the CC, the Maracaibo block, and the EC (reviewed in *Sarmiento-Rojas, 2018*).

From a starting point when shallow seas covered most of the northern Andes 100–66 Ma, three independent lines of evidence suggest that the history of emergence for the EC and the MA is much older than the often cited late Miocene-Pliocene time (5 to 2 Ma, *Gregory-Wodzicki, 2000*). The first and more conspicuous are the widespread and thick, coarse-grained (conglomerate and sandstone) sedimentary sequences of Eocene to Oligocene age (~38–21 Ma) around the EC. Second is the change in sediment sources from the eastern craton, to western, newly raised Andean sources in the EC around ~38 Ma. Both suggest a history of deformation starting at least ~38 Ma. Finally, current global positioning system (GPS) velocities of the northern Andean blocks are more consistent with an older (~38 Ma) mountain building history, than a younger one (5–2 Ma). In the following paragraphs we review each one of these lines of evidence, starting with the Cretaceous history of the northern Andean blocks.

Near the end of the Cretaceous (~66 Ma), oceanic-borne terranes collided with the Cretaceous passive margin of northwestern South America (*Weber et al., 2009*; *Cardona et al., 2012*; *Montes et al., 2019*). Arc-continent collisions built ~2 km of relief along the margin of northwestern South America (*León, Monsalve & Bustamante, 2021*), shedding coarse-grained sand and gravel eastwards, forcing a regional marine regression where swampy conditions were established instead (*Villamil, 1999*; *Gómez et al., 2003*). A primordial, probably discontinuous, CC rose from Guayaquil to the Guajira peninsula, taking ~10 Ma to propagate from south to north, spawning the first Amazonian/Orinoco-style drainages, perhaps shutting down most or all westward-directed drainages of the region (*Hoorn et al., 2010*; *Horton et al., 2010*; *Hurtado et al., 2018*).

Stratigraphic sequences of the EC contain evidence for old deformation (as old as ~66 Ma; *Parra et al., 2012*; *Bayona et al., 2013*; *Mora et al., 2013*), but it is only by ~38–33 Ma

that relief-forming thrust belts propagate into the domain of today's EC (*Gómez et al., 2003*; *Gómez et al., 2005*; *Ochoa et al., 2012*; *Bayona et al., 2013*; *Lamus et al., 2013*). These thrust belts, together with the older CC, define a young Magdalena Valley north of ~4 °N, as a topographic depression between two flanking, linear ranges (CC to the west, and EC to the east, *Horton et al., 2015*; *Reyes-Harker et al., 2015*). The ranges growing in the EC were tall enough to provide potential energy for the erosion and transport of thick, coarse-grained deposits now preserved along both margins of the Magdalena Valley, removing a 2–3 km thick stratigraphic sequence in the process (*Horton et al., 2010*). These ranges were linear, oriented NNE, parallel to similarly oriented thrust faults, constraining also the depositional axis of fluvial strata at that time (*Restrepo-Pace et al., 2004*). South of ~4 °N, the ridges of the young EC would have gently eased (*Mora et al., 2013*) into the lowlands of Amazonia/Orinoco, so the young Magdalena Valley lowlands would have been connected to western Amazonia to the east, and to the Cauca Valley lowlands to the west (*Montes et al., 2021*). To the north, orogenic growth in the Santander Massif (*Ayala et al., 2012*) and Perijá Range (*Montes et al., 2010*) segmented the depositional environments.

By middle Miocene times (Fig. 2, ~15–13 Ma), continued deformation caused widening of the EC (*Mora et al., 2010*), and the MA (*Parnaud et al., 1995*; *Villamil, 1999*; *Bermúdez et al., 2010*) isolating the Maracaibo Basin. At this time the deformation in the EC also began propagating southward (*Saeid et al., 2017*), beginning the isolation of the Magdalena Valley after ~13–11 Ma (*Lundberg & Chernoff, 1992*; *Guerrero, 1997*) with corresponding rock cooling as exhumation took place (*Pérez-Consuegra et al., 2021*). Fossil pollen and spores recovered in ~13 Ma old strata in western Amazonia suggest the presence of high Andean mountain forest in the northern Andes, perhaps as high as ~3 km (*Hoorn et al., 2022*). Final building of the bridge between the EC and the CC (closing the Putumayo corridor, Fig. 1) may have taken place as a result of a combination of further deformation and intense volcanic activity ~3–5 Ma (*van der Wiel, 1991*; *Ujueta, 1999*; *Torres-Hernández, 2010*; *Montes et al., 2021*). At this time (~3–5 Ma) the lowland passes shown in Fig. 1 may have reached their current heights.

Detrital zircon U-Pb geochronology studies the age spectra of ultrastable zircon grains preserved in coarse-grained strata, such as sandstone. This technique provides independent confirmation for the timing of mountain-building summarized above. Cratonic- and Andean-derived clastic loads contain detrital zircon signatures that can be discriminated in vertical stratigraphic successions. Older detrital zircon ages are derived mostly from cratonic sources to the east, while younger ages are mostly derived from Andean sources to the west. Stratigraphic successions east and west of the EC show a marked shift in the transition from Eocene to mid-Oligocene times (~38–28 Ma). In these successions, the age spectra of detrital zircon changes from one dominated by mostly old ages (~540–2,500 Ma coming from the east), to one with additional younger ages (~50–200 Ma, coming from the west, *Horton et al., 2010*; *Nie et al., 2010*; *Bande et al., 2012*). Both of these shifts indicate growth of the EC, erosion, and drainage separation as early as late Eocene times (~38 Ma).

Global Positioning System (GPS) data can be used to determine modern rates and motion paths of crustal blocks in orogenic systems. In the northern Andes, *Mora-Páez*

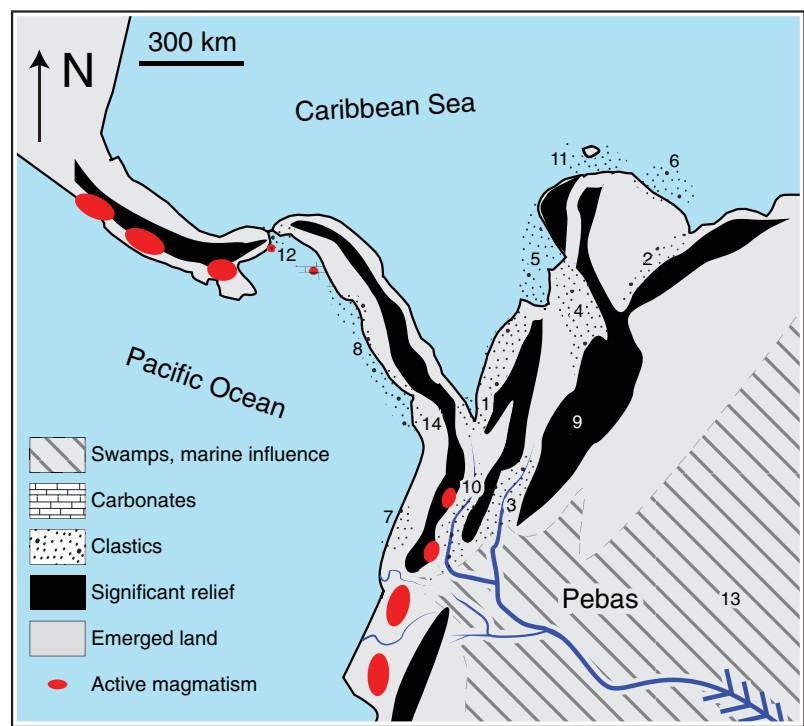

**Figure 2 Paleogeographic reconstruction at middle Miocene times (~13 Ma, modified from *Montes et al., 2021*).** Stippled areas in the figure represent sedimentary environments interpreted from preserved rock sequences; numbers indicate the stratigraphic sequence where it is preserved (1: *Grosse, 1926*; 2: *Parnaud et al., 1995*; *Erikson et al., 2012*; 3: *Guerrero, 1997*; *Montes et al., 2021*; 4: *Gómez et al., 2005*; 5: *Montes et al., 2010*; 6: *Quiroz et al., 2010*; 7: *Borrero et al., 2012*; 8: *Barat et al., 2014*; 9: *van der Hammen, Werner & van Dommelen, 1973*; *Molnar & Pérez-Angel, 2021*; 10: *Echeverri et al., 2015*; *Gallego-Ríos et al., 2020* (10); *Weber et al., 2020*; 11: *Moreno et al., 2015*; 12: *Farris et al., 2017*; 13: *Jaramillo et al., 2017*; *Hoorn et al., 2022*; 14: *León et al., 2018*). Most clastic deposits at this time are fluvial sands and near-shore environments, and mark the segmentation of basins by rising mountain belts and northward-propagating magmatic belts. The absolute elevation of the areas shaded as "significant relief" is unknown, but somewhere in the northern Andes there was proto-paramo vegetation suggestive of ~3 km elevation at this time (*Hoorn et al., 2022*).

*et al. (2016)* show that the rate of orogen-perpendicular (NW-SE) convergence is only ~4 mm/yr. Crustal shortening, that is the total amount of convergence in an orogen, is 100–150 km, and it is usually derived from balanced cross-sections (see compilation in *Montes et al., 2019*). Assuming the convergence rate determined from the GPS data, the time needed to accrue the ~100–150 km would be ~25–38 Ma. This figure is consistent with the evidence reviewed in the previous paragraphs, where thick syntectonic molasse deposits and detrital zircon geochronology also indicate a late Eocene-Oligocene time of initiation of surface uplift, drainage segmentation, and erosion. Assuming a simple, linear velocity of ~4 mm/yr for the last ~25–38 Ma, about half of the surface elevation or higher (*Hoorn et al., 2022*) of the EC would have been reached by ~13–19 Ma, or middle Miocene times.

Contrary to the evidence reviewed in the preceding paragraphs, classic paleobotanical studies of the Bogotá plateau by *van der Hammen, Werner & van Dommelen (1973)*—reviewed and widely cited in the biological literature (*Gregory-Wodzicki, 2000*; *Boschman,*

*2021*)—suggest that about 2,000 m of surface uplift of the EC took place between 5–2 Ma. This paleoelevation figure was derived from palynological changes, which then were converted to temperature ranges (~9–12 °C, (*Hooghiemstra, Wijninga & Cleef, 2006*)), and then to a corresponding elevation change (~2,000 m) according to estimated lapse rates. Published elevation ranges of modern plants, however, may be shown to overlap the paleobotanically derived flora, requiring no elevation change (*Molnar & Pérez-Angel, 2021*). Independently derived paleotemperatures from biomarkers also suggest that the estimated changes in temperature may need to be revised downwards (3 ± 1 °C, *Anderson et al., 2015*). If that is the case, atmospheric cooling—and resulting botanical changes—may simply be the result of changing regional climates in the Miocene-Pliocene transition on an already high orogen (*Pérez-Angel & Molnar, 2017*), consistent with the evidence reviewed above.

In summary, there is now enough geologic data to suggest that the EC grew topographically from north (~6 °N) to south (~3 °N) in the time interval between 38 and 13 Ma. The magnitude of elevation cannot be currently quantified, but extrapolation of current GPS velocity vectors suggest that between 13 and 19 Ma about half of the elevation had been attained. Pollen and spores preserved in the Pebas wetlands in western Amazonia contain high mountain (~3 km) taxa by ~13 Ma, presumably coming from the northern Andes. At this time there still was a direct lowland connection between the Amazon and the Cauca Valley through the Magdalena Valley. Such a lowland connection probably lasted until Pliocene times (~3–5 Ma), when a combination of volcanic and tectonic activity joined the highlands of the EC and the CC and the lowland passes including the Putumayo and Andalucia, reached their current heights (Fig. 1).

## MATERIALS AND METHODS

Collecting permits in Colombia were issued by the *Ministerio de Ambiente y Desarrollo Sostenible* and the *Autoridad Nacional de Licencias Ambientales* (ANLA) in Colombia (*Permiso de estudio con fines de investigación científica en diversidad biológica No. 27 del 22 de junio de 2012, permiso de acceso a los recursos genéticos resolución No. 0377 del 11 marzo de 2014* to AJC and *permiso marco resolución No. 1177* to the Universidad de los Andes). Permits for collecting (#4100: period 2007–2008, and #4750: period 2008–2009, and #4156: period 2009–2010), and for access to genetic resources (#0076 of 22 February 2011) in Venezuela, were issued to FJMRR and MHNLS respectively, by the *Ministerio del Poder Popular para el Ambiente*.

### Species selection and genetic data collection

To re-assess the role of the Eastern Cordillera of Colombia and the Mérida Andes of Venezuelan (EC-MA, Fig. 1) in separating ecologically diverse lowland taxa into eastern and western populations, we combined new information with published data sets representing clades with *cis*- and *trans*-Andean distributions, *i.e.*, east and west of the EC-MA, respectively. We restricted the selection of ingroup samples according to the following filters. First, we included only tetrapods (see Introduction). Second, each set of related populations must comprise a monophyletic group, independently of species names

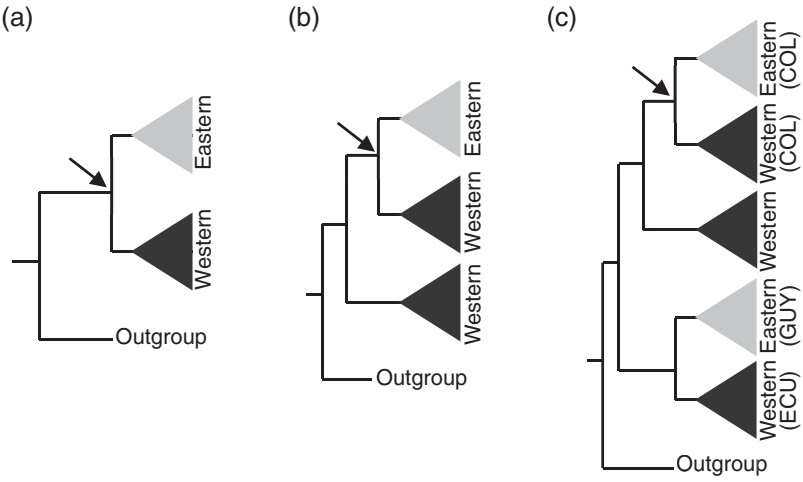

**Figure 3 Assumptions of the comparative phylogeographic analysis in msBayes using hABC (see Methods) required sub-sampling the Bayesian phylogenetic consensus trees.** Arrows mark the subclades selected for analysis by hABC according to the possible tree topology. For reciprocally monophyletic groups with respect to the EC-MA (A) all data were used. For paraphyletic groups (B) we sampled only the clade that fit the two-population model assumed by msBayes. For polyphyletic groups (C) we selected the clade that adjusted to the two-population model with sampling localities geographically closest to the Eastern Cordillera (EC). COL = Colombia, ECU = Ecuador, GUY = Guiana.

currently assigned to the populations. Populations on either side of the EC-MA, however, were not required to be monophyletic to be included in phylogenetic analyses (but see below regarding assumptions of comparative phylogeographic analyses). Third, the chosen clade must include a maximum of one species on at least one side of the EC-MA, while the opposite side could contain the same species, a sister species, or a sister clade with up to a maximum of two named species (Fig. 3). In this way, we sought to limit our study to the most promising potential cases of primary vicariance across the mountains, *i.e.*, the younger splits, regardless of geographic distance of the samples from the EC-MA. Our goal was not to reconstruct the complete history of lineage splitting through time across the Andes, but rather to test specific models, and older divergences could have predated the mountains themselves. Fourth, since we were testing a model of vicariance of lowland taxa by mountain uplift, we excluded tetrapod species whose maximum elevation exceeded 2,000 m. Fifth, in order to exploit coalescent-based analytical tools, we also required data sets to have a minimum of three conspecific individuals sampled on each side of the EC-MA. Sixth, because very few published studies passing the preceding filters also included nuclear DNA data, we were obliged to limit our comparative analyses to mitochondrial DNA (mtDNA) sequence data available in GenBank, in the Barcode of Life Database (BoLD; *Ratnasingham & Hebert, 2007*), or, in the case of frogs, using new data reported here for the first time. We found 32 published data sets meeting the above requirements, and to these we added five previously unpublished data sets from Colombian and Venezuelan frogs. Among these 37 data sets, the total number of lineages per taxonomic class was nine for amphibians, five for non-avian reptiles, 17 for birds, and six for mammals (Table 1). GenBank accession numbers are provided in Table S1.

**Table 1 List of 37 taxa among four taxonomic classes of tetrapods analyzed in the present work, along with abbreviations for the mitochondrial genes used in phylogenetic and comparative phylogeographic analyses.**

| Family | Genus | Species | Gene(s) |
|---|---|---|---|
| Amphibians | | | |
| Hylidae | *Boana* | *B. boans** | COI, 16S |
| Hylidae | *Boana* | *B. xerophylla-platanera** | COI, 16S |
| Hylidae | *Boana* | *B. pugnax** | COI, 16S |
| Hylidae | *Dendropsophus* | *D. microcephalus** | COI, 16S |
| Hylidae | *Scarthyla* | *Sca. vigilans** | COI, 16S |
| Hylidae | *Scinax* | *Sci. ruber** | COI, 16S |
| Leptodactylidae | *Leptodactylus* | *L. bolivianus** | COI |
| Leptodactylidae | *Leptodactylus* | *L. fuscus** | COI, 16S |
| Microhylidae | *Elachistocleis* | *E. ovalis, E. pearsei*, E. bicolor, E. surinamensis* | COI, 16S |
| Reptiles | | | |
| Alligatoridae | *Caiman* | *Ca. crocodilus* | Cytb, COI |
| Boidae | *Boa* | *B. constrictor* | Cytb |
| Colubridae | *Leptodeira* | *L. annulata, L. septentrionalis* | Cytb, ND4 |
| Testudinidae | *Chelonoidis* | *Che. carbonarius* | Cytb |
| Viperidae | *Crotalus* | *Cro. durissus* | Cytb, ND2, ND4 |
| Birds | | | |
| Cardinalidae | *Saltator* | *Sa. maximus* | ND2 |
| Cotingidae | *Querula* | *Q. purpurata* | Cytb |
| Cotingidae | *Schiffornis* | *Sc. turdina* | Cytb |
| Furnariidae | *Automolus* | *A. ochrolaemus* | Cytb |
| Furnariidae | *Glyphorynchus* | *G. spirurus* | Cytb, ND2, ND3 |
| Furnariidae | *Sclerurus* | *Sc. mexicanus* | Cytb |
| Pipridae | *Lepidothrix* | *L. coronata* | Cytb, ND2, ND3 |
| Thamnophilidae | *Cymbilaimus* | *Cy. lineatus* | Cytb |
| Thraupidae | *Chlorophanes* | *Ch. spiza* | Cytb |
| Thraupidae | *Cyanerpes* | *Cy. caeruleus* | Cytb |
| Thraupidae | *Tersina* | *Te. viridis* | Cytb |
| Troglodytidae | *Henicorhina* | *He. leucosticta* | ND2 |
| Troglodytidae | *Microcerculus* | *Mic. marginatus* | Cytb |
| Trogonidae | *Trogon* | *Tro. rufus* | Cytb |
| Tyrannidae | *Colonia* | *Col. colonus* | Cytb |
| Tyrannidae | *Mionectes* | *Mio. oleagineus* | ND2 |
| Vireonidae | *Hylophylus* (now *Tunchiornis*) | *Hy. ochraceiceps* | Cytb |
| Mammals | | | |
| Cebidae | *Cebus* | *Ce. albifrons* | Cytb |
| Cebidae | *Saimiri* | *Sai. sciureus, Sai. oerstedii* | Cytb |
| Didelphidae | *Marmosa* | *Ma. robinsoni* | COI |
| Didelphidae | *Philander* | *P. opossum* | COI |
| Erethizontidae | *Coendou* | *Coe. prehensilis* | Cytb |
| Phyllostomidae | *Trachops* | *Tra. cirrhosus* | COI |

**Note:**
Data sets containing some or all new data published for the first time here are marked with an *. GenBank accession numbers for new and previously published data are provided in Table S1.
## Molecular phylogenetic analyses

To the above sampling we added one or more outgroup samples to each data set in order to root trees and to assist in temporal calibrations of molecular phylogenies. We searched the literature for time-calibrated trees containing the ingroup of interest, and we selected as outgroup the closest species that had published DNA sequences of the same gene or genes as available for the ingroup. Our secondary calibrations assumed previously published divergence time estimates based on fossil data (not on presumed geological events) for amphibians (*Feng et al., 2017*), non-avian reptiles (*Anderson & Greenbaum, 2012*; *Daza et al., 2009*; *Oaks, 2011*; *Vargas-Ramírez, Maran & Fritz, 2010*), and mammals (*Chiou et al., 2011*; *Gutiérrez et al., 2014*; *Hoffmann, Hoofer & Baker, 2008*), while for birds we calibrated based on rates instead of dates (see below). When available, we added additional outgroup species from the same genus or family to reduce uncertainty on node ages. DNA sequence data sets were aligned independently for each genus and each gene using ClustalW v. 1.74 (*Larkin et al., 2007*) with gap opening costs set to 20 for protein-coding genes and 16 for ribosomal RNA genes. PartitionFinder2 (*Lanfear et al., 2012*) was applied independently to each alignment to select the best-fit partitioning schemes and models of nucleotide substitution. Potential partitions included by gene and by codon position.

To estimate divergence times independently for each lineage between eastern and western samples, we employed the Bayesian MCMC molecular phylogenetic software BEAST v. 2.4.7 (*Bouckaert et al., 2014*). We ran two chains for 80 million generations, sampled every 5,000 steps with a coalescent constant-size tree prior, since each data set included population samples and it is the most suitable prior for describing relationships within and between populations. Searches started from a random tree and assumed an uncorrelated lognormal relaxed molecular clock. Data sets with multiple mitochondrial genes were concatenated under the assumption of no recombination in tetrapod mtDNA and assumed substitution models based on PartitionFinder2, as above. For birds, we set the prior on the substitution rate using a lognormal distribution that included the well-established rate for an avian mtDNA molecular clock of 2% total divergence per million years (*Weir & Schluter, 2008*). For anuran mtDNA, we assumed a lognormally distributed substitution rate prior with a mean of 0.00955 divergence per lineage per million years and a range from 0.0074 to 0.01225, corresponding to the 2.5% and 97.5% quantiles, respectively, derived from the silent site divergence rates reported in *Crawford (2003)*. For reptiles and mammals, we employed only secondary calibrations derived from published timetrees (Appendix S1), and we did not constrain substitution rates because the MCMC chains did not converge when we constrained both the node age priors and the substitution rates.

## Comparative phylogeographic analysis

We used the software MTML-msBayes v. 20170511 (*Overcast, Bagley & Hickerson, 2017*) to evaluate the degree of temporal coincidence of divergence events among $n = 37$ tetrapod taxa geographically divided by the EC-MA. MTML-msBayes uses hierarchical approximate Bayesian computation (hABC) to combine data from multiple co-distributed lineages into a global coalescent analysis that includes a hyperparameter, $\Psi$, describing the

number of time intervals necessary to account for the observed range of divergence times among $n$ taxa. Thus a value of $\Psi = 1$ implies simultaneous divergence of all lineages, whereas $\Psi = n$ would mean that $n$ divergence pulses across the EC-MA occurred at a statistically distinct point in time for each taxon (*Huang et al., 2011*). The first step in the hABC analysis is estimating population genetic summary statistics from the observed sequence data. Subsequently, data sets of the same size as the observed data are simulated under a coalescent model using parameter values drawn from a prior distribution, and summary statistics are estimated from each simulated data set. Finally, an acceptance/ rejection algorithm is applied to obtain a sample from the posterior distribution by comparing the summary statistics of each simulated data set with those from the observed data (*Huang et al., 2011*).

For each tetrapod class, we first ran an msBayes analysis with a uniform $\Psi$ prior that ranged from 1 to the maximum number of possible divergences. Then we employed the estimated dispersion index Omega ($\Omega = \text{var}[\tau])/E[\tau]$, where $\tau$ is the age of the divergence pulse that divided the ancestral population and $E[\tau]$ is the mean divergence time, to estimate the relative support of the hypothesis of a synchronous divergence pulse. Thus, when $\Omega$ is close to 0 a single divergence pulse is supported, whereas a greater value of $\Omega$ implies multiple divergence pulses. We calculated Bayes Factors (BF; *Kass & Raftery, 1995*) to compare the posterior strength of synchronous and asynchronous divergence models. We defined four $\Omega$ thresholds (0.01, 0.025, 0.05 and 0.1) to calculate $BF_{01}$ ($M_0$ = Synchronous divergence, $\Omega \leq$ threshold; $M_1$ = Asynchronous divergence, $\Omega >$ threshold). We assumed that mitochondrial genes are in perfect linkage disequilibrium, and therefore treated multi-gene data sets as single, concatenated sequences in hABC, and excluded any sample with more than 50% of missing data. MTML-msBayes assumes a two-population model (Fig. 3A) that is violated by deep phylogeographic structure on one or both sides of the proposed barrier (*Huang et al., 2011*). Therefore, we checked tree topologies obtained using BEAST (Appendix S2). We employed all samples from the phylogenies that fit the assumptions of the MTML-msBayes model, whereas for paraphyletic and polyphyletic groups we included only samples from the clade that best fit the two-population model. Thus, we included the clade with the most samples (to obtain better estimates of population genetic parameters) and/or clades geographically positioned most closely to the EC-MA (Figs. 3B and 3C; Appendix S2), and therefore samples in Fig. 1 located far from the EC-MA were not genetically more distant. In some cases, one or two samples from one population were nested inside the clade of the opposite flank of the EC-MA. Whether caused by migration or by incomplete lineage sorting, such paraphyly violated the historical demographic model assumed by MTML-msBayes, and we were obliged to remove those one or two samples from our analyses (M. Hickerson, 2018, personal communication).

We ran one independent analysis for each of the four classes of tetrapods. We set the upper limit of $\tau$ to the oldest mean split age for each class according to our BEAST analysis. Divergence times estimated from MTML-msBayes are in coalescent units, thus the conversion to millions of years assumed roughly equal sex ratios, haploid and maternally inherited mtDNA, and was made following the equation $t = \tau \theta_{\text{Ave}}/\mu$, where $\mu$ is the neutral

mutation rate per site per generation and $\theta_{\mathrm{Ave}}$ is the product of $\mu$ and the average effective population size for each taxonomic class estimated by MTML-msBayes. Values of $\mu$ for frogs were taken from *Crawford (2003)* and for birds we followed *Dolman & Joseph (2012)*. For mammals and non-avian reptiles, we used specific rates per genus. Further details regarding estimating coalescent-based divergence times, hyperpriors, substitution rates, and generation times assumed for each taxon are given in Appendix S3. Hyper-posteriors were estimated from 1,000 accepted draws from 1.5 million simulations. We made a local linear regression of the accepted parameter values obtained by the acceptance/rejection step in order to improve the posterior estimation.

## Limitations of single locus mtDNA

We employed mtDNA to infer phylogenetic relationships and the number and the timing of divergence pulses. This marker is widely used in comparative phylogeographic studies for several reasons. Its high mutation rate and lack of recombination provide ample and detailed information needed to reconstruct the genealogy of samples with more precision than found for any other single marker (although it can show bias relative to the nuclear genome). The fact that it is haploid and maternally inherited means genetic drift is at least 4-fold greater, which greatly lowers the probability of incomplete lineage sorting (ILS; *Avise et al., 1987*; *Hudson & Coyne, 2002*). Finally, the conserved structure of the mitochondrial genome makes it the best single marker for comparative analyses across diverse animal species (*Carnaval et al., 2009*), though mtDNA does not always predict variation in nuclear DNA (nDNA; *Toews & Brelsford, 2012*). In some studies, discordances between mtDNA and nDNA phylogenies are observed, so analyses including different loci are recommended to confirm phylogeographic histories. We are aware of the importance of nDNA for modern phylogeographic analyses, especially when studying lineages that show some migration or hybridization. Nonetheless, we consider that this limitation is compensated for by the ample taxonomic coverage of our study, made possible through the extensive and well-documented public mtDNA data sets. Furthermore, the approach used in hABC takes into account the intrinsic stochasticity in single-locus coalescence genealogies across different taxa, and estimation of population divergence time ($\tau$) does not improve considerably until eight nuclear loci are incorporated (*Huang et al., 2011*). Unfortunately, there are no public data sets with that degree of genetic sampling available for lowland tetrapods located east and west of the EC-MA.

## Life-history determinants of divergence times

We selected six life-history variables potentially related to dispersal abilities that could influence divergence times: body size as length (we could not find mass data for all species), geographic range, upper elevation limit, type of locomotion (flying *vs.* non-flying), thermoregulation (homeotherm *vs.* poikilotherm) and taxonomic class (amphibians, non-avian reptiles, birds, mammals). We obtained snout-vent length of amphibians from the AmphiBIO database (*Oliveira et al., 2017*). For reptiles we obtained body-size data from the literature (*Savage, 2002*; *Bartlett & Bartlett, 2003*; *Böhm et al., 2013*; *Fowler, 2018*). We obtained body-size data for birds from the *Handbook of the Birds of the World* (*del*

**Table 2 List of the thirteen models selected *a priori* that could potentially explain differences in divergence times among data sets, the biological relevance of each model and their corrected Akaike Information Criterion (AICc) coefficients.**

| Model | Biological relevance | AICc | ΔAICc |
|---|---|---|---|
| Temperature regulation | Homeotherms can tolerate environmental gradients, facilitating dispersal. | 88.09 | 0 |
| Upper elevation limit × Temperature regulation | Homeotherms can tolerate lower temperatures associated with elevation gradients and thus could reach higher elevations. | 91.16 | 3.07 |
| Upper elevation limit | Physical barriers would have greater impact for species with lower elevation limits. | 91.59 | 3.5 |
| Class | Differences in divergence times depend only on taxonomy. | 92.13 | 4.04 |
| Geographic range + Upper elevation limit | Species with wide geographic and elevation ranges could tolerate environmental heterogeneity | 93.14 | 5.05 |
| Dispersal | Flying organisms would be better dispersers. | 93.25 | 5.16 |
| Body size | It is easier for larger organisms to disperse. | 93.62 | 5.53 |
| Geographic range | Organisms with wider geographic ranges can be considered generalists and thus better dispersers. | 93.72 | 5.63 |
| Body size + Geographic range + Upper elevation limit | Large species with wide geographic ranges and higher elevation limits could better tolerate environmental heterogeneity and would be more able to disperse. | 95.23 | 7.14 |
| Body size + Geographic range | Large and generalist species could disperse longer distances. | 95.74 | 7.65 |
| Body size × Class | Body size can vary across tetrapod classes, influencing dispersal abilities. | 99.74 | 11.65 |
| Upper elevation limit × Class | Elevation limit could differ among classes. | 99.89 | 11.8 |
| Geographic range × Class | Geographic range could vary depending on the taxonomic class of the organism. | 100.46 | 12.37 |

**Note:**
Each model contained one, two, or three 'life history' variables. Linear combinations of variables are represented by a '+' symbol, while linear combinations that also include an interaction between variables are represented by '×'.

*Hoyo et al., 2017*), and for mammals we consulted the amniote life-history database (*Myhrvold et al., 2015*). We used species distribution shapefiles from the *IUCN (2019)* to estimate the geographic ranges of species in km$^2$. We obtained upper elevational-limit data from the *IUCN (2019)*, Amphibian Species of the World (*Frost, 2019*), and the *Handbook of the Birds of the World*. These life history and geographic range data are available in Table S1 under the tab 'GLM Data'. The variables mean divergence time and body size were transformed with the natural logarithm function to better meet assumptions of normally distributed residuals and homoscedasticity. With only $n = 37$ data points to evaluate the above six life-history variables, we defined a total of 13 *a priori* general linear models (GLM; models explained in Table 2) that could potentially explain variation in divergence times, and performed an AICc model selection procedure using R v. 3.5.1 (*R Core Team, 2018*) to determine which variables were more relevant in determining divergence times.

## Data availability
All DNA sequence data analyzed in this work are available in GenBank, including new DNA sequence data from frogs generated for this study. All GenBank accession numbers are provided in Table S1. DNA sequence alignments for each taxon are provided as NEXUS files on FigShare, and the URL for each file is provided in the final tab of Excel file Table S1.

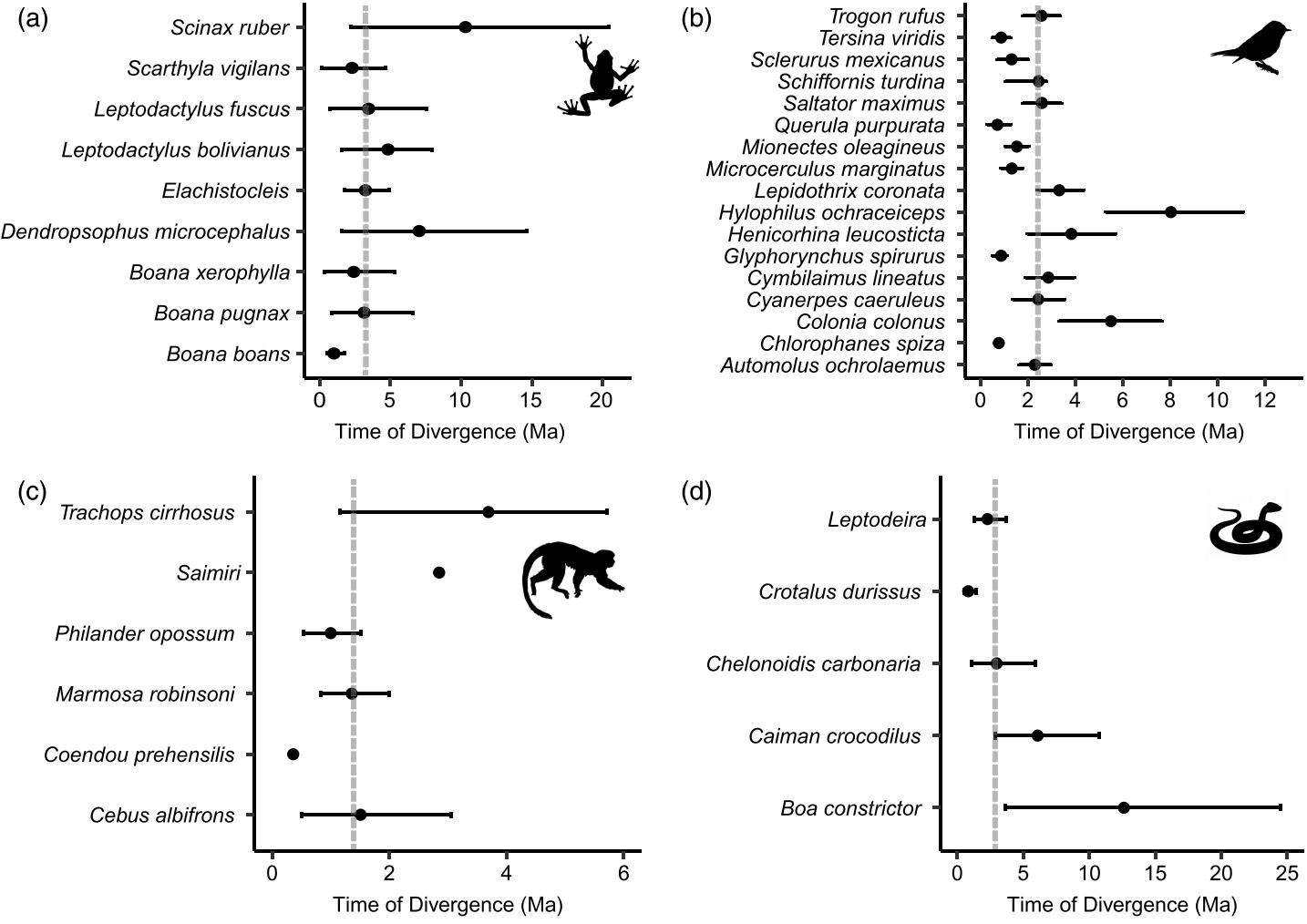

**Figure 4 Distribution of divergence time estimates between eastern and western lineages of (A) frogs, (B) birds, (C) mammals and (D) non-avian reptiles, estimated by Bayesian MCMC relaxed-clock phylogenetic analysis of mitochondrial DNA.** Dots indicate the mean divergence time and dotted lines indicate the median divergence time within each class. Median divergence time in frogs was 3.24 million years ago (Ma), 2.99 Ma in non-avian reptiles, 2.44 Ma in birds, and 1.44 Ma in mammals.

## RESULTS

### Divergence time estimation and hABC

All dated molecular phylogenies generated by BEAST, with relevant nodes of interest highlighted, are available in Appendix S2. For frogs, the age of the node that split east and west populations in the BEAST analyses ranged from 10.3 to 1.01 Ma (median 3.24 Ma; Fig. 4A), for birds 8.04 to 0.72 Ma (median 2.44 Ma; Fig. 4B), from 12.62 to 0.84 Ma in non-avian reptiles (median 2.99 Ma; Fig. 4C), and from 3.69 to 0.36 Ma among lineages of mammals (median 1.44 Ma; Fig. 4D). Using MTML-msBayes, the estimated $BF_{01}$ among all thresholds strongly supported a scenario of asynchronous divergence in amphibians, in non-avian reptiles, and in birds (Table 3), whereas the estimated $BF_{01}$ for mammals suggested weak evidence of synchronous divergence. The support for synchronous divergence increased when the threshold of $\Omega$ (the dispersion index of all divergence

**Table 3 Estimated Bayes Factors comparing models of synchronous and asynchronous divergence ($BF_{01}$; $M_0$ = Synchronous divergence, $M_1$ = asynchronous divergence).**

| Threshold | $BF_{01}$ | Strength of evidence |
|---|---|---|
| Amphibians | | |
| 0.01 | 0.006 | Strong support for $M_1$ |
| 0.025 | 0.011 | Strong support for $M_1$ |
| 0.05 | 0.013 | Strong support for $M_1$ |
| 0.1 | 0.015 | Strong support for $M_1$ |
| Reptiles | | |
| 0.01 | 0.003 | Strong support for $M_1$ |
| 0.025 | 0.003 | Strong support for $M_1$ |
| 0.05 | 0.003 | Strong support for $M_1$ |
| 0.1 | 0.012 | Strong support for $M_1$ |
| Birds | | |
| 0.01 | 0.014 | Strong support for $M_1$ |
| 0.025 | 0.025 | Strong support for $M_1$ |
| 0.05 | 0.025 | Strong support for $M_1$ |
| 0.1 | 0.029 | Strong support for $M_1$ |
| Mammals | | |
| 0.01 | 0.964 | Anecdotal evidence for $M_0$ |
| 0.025 | 1.054 | Substantial evidence for $M_0$ |
| 0.05 | 1.344 | Substantial evidence for $M_0$ |
| 0.1 | 2.045 | Substantial evidence for $M_0$ |

**Note:**
$BF_{01} > 100$ supports strongly $M_0$, whereas $BF_{01} < 1/100$ supports strongly $M_1$. Note, MTML-msBayes was necessarily run for each taxonomic class independently (see Methods).

pulses) was higher. The timing of intervals of divergence events estimated by coalescent analysis and the participating species within each interval are given in Table 4.

To evaluate the potential congruence in divergence time estimates between Bayesian MCMC phylogenetic inference and coalescent methods implemented in MTML-msBayes, we compared the node ages estimated by BEAST against the average pairwise net difference between eastern and western populations, $\pi_{net}$ (Appendix S4). MTML-msBayes uses $\pi_{net}$ to estimate which pair of populations belong to each divergence pulse, so it is expected that taxa with greater $\pi_{net}$ would present older divergences. In general, we found a positive tendency between node ages and $\pi_{net}$ (Appendix S4; correlation coefficient r in amphibians r = 0.44; reptiles r = 0.86; birds r = 0.53; mammals r = 0.28). However, we found a few pairs of populations that did not followed the tendency, showing low $\pi_{net}$ values and old divergence times, namely *Dendropsophus microcephalus, Scinax ruber* (frogs); *Hylophilus* (now *Tunchiornis*) *ochraceiceps* (birds); *Saimiri* and *Trachops cirrhosus* (mammals). We also found population pairs with recent divergences and high $\pi_{net}$ values: *Sclerurus mexicanus* and *Trogon rufus* (birds). These discrepancies could be attributed to the parameters we employed to run each test. BEAST estimates a timetree assuming complex models of substitution and here we also constrained the age of at least one basal

**Table 4 Number, grouping, and timing of each divergence interval (Ψ) estimated by hierarchical approximate Bayesian computation (hABC) as implemented in the software, MTML-msBayes.**

| Number of intervals (Ψ) | Species within temporal interval | Time (Ma) | 95% CI in Ma |
|---|---|---|---|
| Amphibians | | | |
| $\Psi_1$ | *B. boans, Sca. vigilans* | 0.13 | [0–0.45] |
| $\Psi_2$ | *B. xerophylla-platanera, Elachistocleis, L. fuscus* | 2.14 | [0.85–3.5] |
| $\Psi_3$ | *B. pugnax, D. microcephalus* | 3.62 | [1.97–5.83] |
| $\Psi_4$ | *L. bolivianus, Sci. ruber* | 7.57 | [6.29–8.69] |
| Reptiles | | | |
| $\Psi_1$ | *Cr. durissus, Che. carbonarius* | 1.9 | [0.51–3.02] |
| $\Psi_2$ | *Ca. crocodilus* | 4.59 | [1.5–8.66] |
| $\Psi_3$ | *B. constrictor, Leptodeira* | 9.05 | [6.3–11.8] |
| Birds | | | |
| $\Psi_1$ | *Chl. spiza, Q. purpurata, Scl. mexicanus, Te. viridis* | 0.23 | [0–0.47] |
| $\Psi_2$ | *Cya. caeruleus, Cym. lineatus, Mio. oleagineus* | 0.65 | [0–1.2] |
| $\Psi_3$ | *Col. colonus, Le. coronata, Sa. maximus* | 1.53 | [0.65–2.33] |
| $\Psi_4$ | *G. spirurus, He. leucosticta, Sch. turdina* | 2.10 | [1.22–2.96] |
| $\Psi_5$ | *A. ochrolaemus, Mic. marginatus, Tro. rufus* | 2.67 | [1.62–4.06] |
| $\Psi_6$ | *Hy. ochraceiceps* | 7.73 | [6.07–9.44] |
| Mammals | | | |
| $\Psi_1$ | *Ce. albifrons, Co. prehensilis, Ma. robinsoni, P. opossum, Saimiri, Tra. cirrhosus* | 0.36 | [0.31–0.43] |

**Note:**
Intervals are independently estimated within each of the four taxonomic classes. Complete generic level names are found in Table 1. Ma = millions of years ago. CI = posterior credibility interval.

node (except in birds). Moreover, MTML-msBayes uses a coalescent approach in which only substitution rates are needed and there is only one substitution model available for our type of data. Because our phylogenetic analyses in BEAST assumed substitution rates per million years (not per generation) and the divergence asynchrony analyses were based only on $\pi_{net}$, any uncertainty in generation time would affect only the divergence times estimates based on MTML-msBayes (see Appendix S3 for the generation time assumed for each taxon).

## Life-history determinants of divergence times

Among the thirteen models we evaluated to explain variation among species in divergence times estimated by BEAST, the highest ranked model contained only a single eco-physiological variable, 'thermoregulation' (AICc = 91.28; ΔAICc ≡ 0; Table 2), with homeothermic species showing younger divergence times relative to poikilotherms (Fig. 5; mean node age in homeotherms = 2.35 Ma; poikilotherms = 4.46 Ma). However, this model explained only a small portion of the variation ($p = 0.02$; $R^2 = 0.14$; adjusted $R^2 = 0.12$). The second best model explained a little more of the variation and contained the variables thermoregulation and upper elevation limit (ΔAICc = 3.07; $R^2 = 0.19$; adjusted $R^2 = 0.12$; Table 2). Taxonomic class appeared to be the least informative predictor of divergence across the EC-MA since models including this variable as an additive effect or as an interaction term had the worst AICc values: class and elevation

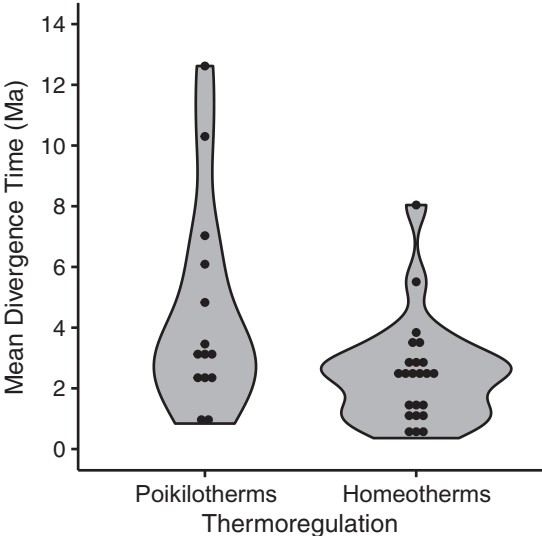

**Figure 5 Mean divergence times (in million of years ago, Ma) of eastern and western populations as estimated by Bayesian molecular phylogenetic inference of mitochondrial DNA sequence data for homeotherm and poikilotherm tetrapods.** The points represent the mean divergence time of each of 37 data sets (GLM, $df = 35$, $t = −2.43$, $p = 0.02$, adjusted $R^2 = 0.12$; mean node age in poikilotherms = 2.35 Ma; mean node age in homeotherms = 4.46 Ma).

($df = 29$; $F = 1.57$; $p = 0.18$), class and body size ($df = 29$; $F = 1.59$; $p = 0.17$) or class and range size ($df = 29$; $F = 1.48$; $p = 0.21$).

## DISCUSSION

### Pulses of divergence and Andean uplift

Comparison of phylogeographic patterns among co-distributed lineages allows the researcher to test the role of geological events and landscape features as possible determinants of speciation that shaped current geographic patterns of biodiversity (*Hickerson et al., 2010*). The typical null hypothesis assumes that co-distributed species share a common biogeographic history influenced by the same historical events (*Arbogast & Kenagy, 2008*). We reject our null hypothesis H0, as our results demonstrate the occurrence of several divergence pulses. First, mean Ω (the dispersal index of all divergence pulses) in amphibians, non-avian reptiles and birds were higher than the proposed thresholds, implying asynchronous divergence. Second, estimated divergence times, τ, based on a coalescent model vary so widely among lineages within taxonomic classes of tetrapods that no single time interval could account for this temporal variation.

The amphibian, reptile, and bird data sets each have at least one divergence pulse in which the mean τ is older or younger than the oft-cited age of EC-MA uplift (5–2 Ma; Table 4). Only in mammals did the hABC framework support a single divergence time interval. However, at 0.43–0.31 Ma, this interval of divergence ages is far younger than any model for orogeny of the EC-MA (Table 4), even the final formation of the lowest mountain passes. Thus, the variation in divergence times among taxa, as well as the number of splits outside the traditional 5–2 Ma interval, suggest we need to reconsider our biogeographic
models of the effect of uplift of the EC-MA mountain chain on the diversification of lowland species.

## Two alternative scenarios to Andean vicariance

Our synthesis of recent geological studies (Fig. 2) combined with our comparative phylogeographic analyses failed for three reasons to support the traditional model of vicariance mediated by the uplift of the northern Andes. Geological evidence suggests the EC-MA had already reached significant elevation as early as 38 Ma north of ~4 °N, at least 3 km by 13 Ma (*Hoorn et al., 2022*), and the last mountain passes (Fig. 1), would have been complete at ~2 °N, by 5–3 Ma (see below). Coalescent analyses show that divergence was asynchronous within taxonomic classes, except for mammals, as well as among classes. Time-calibrated phylogenetic analyses, with some notable exceptions, revealed very young divergences and even paraphyly and polyphyly of mtDNA lineages across the EC-MA. Here we consider potential explanations for variation among our 37 lowland clades in divergence times across the EC-MA, coupled with the trend towards Plio-Pleistocene ages. Under a model of pure vicariance caused by a physical barrier, populations on either side start as polyphyletic entities that, in the absence of subsequent migration, will reach reciprocal monophyly at a rate inversely proportional to effective population size (*Avise et al., 1983*), such that monophyly would be reached four times faster in mtDNA than in nDNA (*Hudson & Coyne, 2002*) or even faster (*Crawford, 2003*). Of the 37 tetrapod lineages studied here, 17 (46%; three amphibians, three non-avian reptiles, nine birds, and two mammals) showed polyphyly with respect to the position of the EC-MA. If this frequent polyphyly is due to incomplete lineage sorting, then this provides further evidence that the hypothesized vicariant split was recent relative to population size, and would suggest that the potential impact of the EC-MA on some lowland tetrapods took place much more recently than traditionally accepted ages for the end of the uplift of the EC-MA. The asynchrony in timing of divergence makes this explanation less likely, however. Alternatively, polyphyly could be created by multiple crossings of the EC following an initial vicariant separation, as found in, for example, the bird *Mionectes* (*Miller et al., 2008*) and the mid-elevation frog, *Rheobates* (*Muñoz-Ortiz et al., 2015*). *Smith et al. (2014)* demonstrated that genetic patterns in cross-Andean birds are not explained by a simple vicariant scenario and that the Andean cordillera was not a driver of vicariance but acted as a semi-permeable barrier that allowed recent dispersal and that diversification was driven by the capacity of these birds to disperse. This trend towards Plio-Pleistocene divergences coupled with a heterogeneous response to landscape features has also been reported among Lower Central American phylogeographic studies (*Bagley & Johnson, 2014*). The temporal patterns found in our results confirm that the EC-MA is an ineffective barrier for at least half of tetrapod lineages studied here including several amphibians and reptiles. Such a model of recent dispersal across a fully formed Andes could explain the heterogeneity in divergence times among lineages, the overall young ages relative to the old age of the EC-MA as synthesized here, and the young divergence times among homeotherms relative to poikilotherms (Fig. 5).

The main alternative to the cross-Andean dispersal scenario outlined above would be dispersal through lowland passes. In the Present we can observe three mountain passes that potentially could connect lowland populations to the east (Amazonia and Llanos) and west (Magdalena and Maracaibo watersheds) of the EC-MA. From south to north they are the Putumayo corridor, Andalucía pass, and Táchira corridor (Fig. 1). The lowest of these passes is 1,900 m elevation, however, and our lowland data set did not knowingly include any species with locality records of 2,000 m or above, suggesting that these passes are currently 'closed' to most, if not all, of our sampled taxa. Paleontological evidence, however, indicates that these passes may have been far lower during the Miocene. The Magdalena River valley (Fig. 1) contains fossil fishes of Amazonian affinities, suggesting that a very low pass through the southern EC was open until at least 11 Ma (*Lundberg & Chernoff, 1992*). Fossil catfishes described more recently from northern Venezuela suggest a very low pass existed through the MA as recently as 5–3 Ma (*Aguilera et al., 2013*). Two possible lowland passes through an otherwise tall and continuous EC-MA may have existed. The western end of the MA could have connected Amazonia and the Caribbean coast to the north as recently as ~3 Ma, perhaps corresponding to the Táchira corridor. The southern end of the Magdalena Valley may have allowed lowland connectivity between Amazonia and the Magdalena from ~13 until ~5 Ma (*Montes et al., 2021*), corresponding to the Putumayo corridor or Andalucía pass in Fig. 1 (*Kroonenberg et al., 1981*; *van der Wiel, 1991*; *Ujueta, 1999*). Using phylogeographic analyses to testing for the imprint of a specific pass across the southern EC may prove challenging, however, because the environmental conditions today differ dramatically between the two sides, with semi-arid conditions on the Magdalena River side contrasting sharply with the humid Amazonian moist forest to the east, and very few if any species are found on both sides.

## Ecological factors and divergence times

The only life-history variable that was found to be associated with divergence times was thermal physiology, such that poikilotherms had older divergences across the EC-MA than homeotherms (mean node age in poikilotherms and homeotherms were 4.46 Ma and 2.35 Ma, respectively), supporting hypothesis H3A. We found no support for H1 looking at species' ranges, H2 involving body size, nor for H3B concerning locomotion, but also recognize that our analysis may have lacked power with only $n = 37$ lineages. Poikilotherm dispersal is restricted by temperature and these animals cannot perform across as wide a range of environmental temperatures as homeotherms can (*Porter & Gates, 1969*). Thermal biology may explain why amphibian and reptile species tend to show Miocene to Pliocene intraspecific divergences (crown ages), whereas mammals and birds tend to show late Pliocene to Pleistocene divergences (*Turchetto-Zolet et al., 2013*). While the $R^2$ was low for thermal biology in our analysis, this factor was still more important than body size or even type of locomotion (flying *vs*. non-flying). Tropical lowland poikilotherms have restricted elevational ranges, narrow temperature tolerances and limited acclimation responses, as compared to high-elevation or temperate poikilotherms species (*Ghalambor et al., 2006*; *Sheldon et al., 2018*; *Pintanel et al., 2019*), although in our analyses, neither

upper elevation limit nor geographic range were significant predictors of divergence time across the EC-MA.

The significant albeit weak relationship between divergence times and thermal biology (Fig. 5) provides a clue that the interaction between life history and environment may have played a role in structuring variation across the Andes, as has been found in a related comparative phylogeographic study of frogs in Panama (*Paz et al., 2015*). While the EC-MA was largely formed in the late Eocene times (38–33 Ma), global temperature began to decrease from this epoch and became more abrupt during late Miocene and Pleistocene (*Flantua et al., 2019*; *Muellner-Riehl et al., 2019*). In highland ecosystems of the EC, Pleistocene climate oscillations created a 'flickering connectivity', such that during glaciation events the paramo ecosystems were more connected, while during interglacial periods they returned to being more isolated on mountain top (*Flantua et al., 2019*). Lowland ecosystems may have experienced the opposite pattern, such that when temperatures increased during interglacials, lowland taxa would have moved upwards in elevation (*Flantua & Hooghiemstra, 2018*; *Hooghiemstra & Flantua, 2019*). Interglacial periods may have favoured dispersal over the EC-MA, although temperatures may not have ever been as high as they are now. Connectivity promoted by climate change would likely favour homeotherms relative to poikilotherms since the former can better withstand lower temperatures and thus disperse more easily across an elevational gradient. Thus, environmental fluctuations may have been an additional factor driving recent patterns of genetic variation across the EC-MA in some lowland tetrapods, and our two alternative scenarios to Andean vicariance may have been operating at the same time, such that climatic fluctuations may have facilitated the use of previously lower mountain passes by the ancestral lowland populations east or west of the EC-MA.

In summary, we integrated new and previously published lines of evidence here that we hope can be accounted for by one or two historical explanations: (1) geological evidence revealing old age of uplift for the EC-MA in general dating to around 38–33 Ma, coupled with (2) paleontological evidence from Amazonian lineages of fishes found in the southern Magdalena Valley at 13–5 Ma and northern Venezuela at 5–3 Ma. (3), asynchrony (aka, high variance) in divergence times (coalescent estimation using hABC) within and among groups of tetrapods, ranging from 0.13 Ma in *Boana boans* to 9.05 Ma in *Boa constrictor*, and, finally (4) low median age of divergence (gene-tree estimates using BEAST) within each tetrapod group from 3.26 Ma in amphibians to 1.43 Ma in mammals. Because a traditional model of Andean uplift cannot account for the young ages or the variation in divergence times among tetrapod lineages spanning the northern Andes, we propose that a portal in the southern EC, potentially interacting with Pleistocene climate fluctuations, explain the geological and genetic data much better.

## CONCLUSION

Geology and evolutionary genetics offer complementary views of the history of our planet. Any historical hypothesis generated from genetic data should be evaluated with geological information, and vice versa, in addition to incorporating available paleoclimatic and paleontological data. The present analysis is unique in explicitly synthesizing geological

and genetic data and perspectives, including one of the first comparative studies involving organisms with widely varied physiological and ecological traits, to evaluate the role of the northern end of the World's longest mountain chain, the Andes of South America, in promoting lowland diversification. We find that, while vicariance likely played a role for some taxa, especially the older lineages, our synthesis of all available data gives more support for frequent and recent dispersal across the EC-MA. Recent divergences and polyphyly of gene trees demonstrate that the EC-MA did not limit dispersal completely even posterior to final uplift. Because we focused on lowland taxa, we conclude that such dispersal may have been facilitated by a trans-Andean portal and/or Pleistocene climate fluctuations. We look forward to further testing these hypotheses using a greater diversity of lineages and genetic markers.

## ACKNOWLEDGEMENTS

We are grateful to Michael J. Hickerson for providing feedback and recommendations for running MTML-msBayes.

### Funding

This project was funded by the Facultad de Ciencias of the Universidad de los Andes under the call Proyectos cortos o generación de un producto adicional, and DIDI (Dirección de Investigación, Desarrollo e Innovación) of the Universidad del Norte under project "El nacimiento del Magdalena: sus ancestros de Amazonas y el Orinoco". New DNA sequence data reported here for frogs were obtained with the generous support of a grant in Basic Sciences No. 360-2013 from Colombia's Colciencias (now the Ministerio de Ciencia Tecnología e Innovación) to Andrew J Crawford. The funders had no role in study design, data collection and analysis, decision to publish, or preparation of the manuscript.

### Grant Disclosures

The following grant information was disclosed by the authors:
Universidad de los Andes.
DIDI (Dirección de Investigación, Desarrollo e Innovación).
Ministerio de Ciencia Tecnología e Innovación: 360-2013.

### Competing Interests

The authors declare that they have no competing interests.

### Author Contributions

- Erika Rodriguez-Muñoz conceived and designed the experiments, performed the experiments, analyzed the data, prepared figures and/or tables, authored or reviewed drafts of the article, and approved the final draft.
- Camilo Montes conceived and designed the experiments, prepared figures and/or tables, authored or reviewed drafts of the article, and approved the final draft.

- Fernando JM Rojas-Runjaic conceived and designed the experiments, performed the experiments, analyzed the data, prepared figures and/or tables, authored or reviewed drafts of the article, and approved the final draft.
- Andrew J. Crawford conceived and designed the experiments, performed the experiments, analyzed the data, prepared figures and/or tables, authored or reviewed drafts of the article, and approved the final draft.

## Animal Ethics

The following information was supplied relating to ethical approvals (*i.e.*, approving body and any reference numbers):

Collecting permits in Colombia were approved by the Ministerio de Ambiente y Desarrollo Sostenible and the Autoridad Nacional de Licencias Ambientales (ANLA) in Colombia (Permiso de estudio con fines de investigación científica en diversidad biológica No. 27 del 22 de junio de 2012, permiso de acceso a los recursos genéticos resolución No. 0377 del 11 marzo de 2014 to AJC and permiso marco resolución No. 1177 to the Universidad de los Andes). Permits for collecting (#4100: period 2007–2008, and #4750: period 2008–2009, and #4156: period 2009–2010), and for access to genetic resources (#0076 of 22 February 2011) in Venezuela, were issued to FJMRR and MHNLS respectively, by the Ministerio del Poder Popular para el Ambiente.

## Data Availability

The sequences and accession numbers are available in Table S1.

## Supplemental Information

Supplemental information for this article can be found online at http://dx.doi.org/10.7717/peerj.13186#supplemental-information.

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
