# Peer review of "Synthesis of geological data and comparative phylogeography of lowland tetrapods suggests recent dispersal through lowland portals crossing the Eastern Andean Cordillera"

_PeerJ, doi:10.7717/peerj.13186_

## Round 0.1 · original submission · Minor Revisions

I have now received four reviews for your paper and they all agree that your study is well designed, written, and worthy of publication, however, with some revision. I agree with this endorsement. The comments of all the reviewers are encouraging, constructive, reasonable and self-explanatory and hence please address these as much as possible in your revision.

Reviewer 1 ·

Basic reporting

these are all suitable.

Experimental design

Methods are adequately described and hypotheses are clearly stated.

Validity of the findings

My only concern is how far away some sample localities are from the cordillera (see Additional comments).

Additional comments

This manuscript uses divergence ages of published and new mtDNA from 37 tetrapods across the northern Andes along with a summary of geological evidence regarding uplift of this region to understand whether lineages diverged due to simple vicariance. I do not have many comments, and I think this manuscript serves an important role in emphasizing the oversimplification that often happens with vicariant studies in orogenic contexts. I do think the geology portion will be too detailed for many biologists, but I think the detail is necessary to the aim of the paper and to reconstruct the dual biotic and abiotic histories.
I suggest that the authors consider how different generation times would impact the distribution of their ages (i.e. a range of divergence ages would be expected as a function of generation time). Finally, regarding lines L542-558, were these passes always low or did they become low at this time? Given the importance of uplift history that the authors review early on, it seems surprising to mention only in the discussion that there are low passes that could have accommodated dispersal all along?

Figure 1 – (i) is it possible to increase the hillshade on this to show topography? (ii) if these are sample localities for the genetic data on the inset, is it a concern that some of the sampling points are extremely far away from the cordillera?

Figure 2 – I think for this figure to be useful to biologists it requires more explanation as to how it relates to this study. Adding the ages for middle Miocene to the figure itself would also help as well as defining the subduction zone/fault lines.

Reviewer 2 ·

Basic reporting

This is a very nice multi-taxon approach to vicariance across the Andes. We need more such comparative studies to make broad biogeographic statements. The main limitation is that it is based entirely on mtDNA, which can “reset” divergence times through subsequent introgression, but the authors fully appreciate this fact and comment on it. The paper is very well-written and referenced, with the possible exception of reviews on Andean phylogeographic studies.

52-54 My interpretation has always been that mountains are not so much a physical barrier, but a habitat break (i.e. alpine splitting montane or lowland distributions), just as the tropics isolate (cool)temperate species in each hemisphere.

156-263 This section on geology is too long and detailed for readers. It would be better to focus only on geology as it relates to specific questions, when they are defined, and only give a brief “geological setting” in the Intro. Could it be reduced by using Figures?

In general, there is minimal reference to phylogeographic studies on the Andes. Mention more reviews of studies of the area.

389 It is more usual to talk about drift being “greater”, not “faster”, since it is a stochastic dispersive process. Fixation might be faster- yes.

517-18 Why mention nDNA again when it is not relevant for this paper? You’ve already said this earlier anyway.

Experimental design

The main limitation is that it is based entirely on mtDNA, which can “reset” divergence times through subsequent introgression, but the authors fully appreciate this fact and comment on it.

Validity of the findings

No comment

Additional comments

No comment

·

Basic reporting

This is an extremely well written article that uses cutting-edge methods to reach important and general conclusions. I have an uncharacteristically short list of criticisms. (1) Given the number of taxa and trees, it would be useful to post the alignments (not simply the Genbank accession numbers) on a public repository (e.g. Figshare); (2) given the large number of sequences downloaded from public genetic repositories, it would help orient the reader to the extent of the authors' own sequencing efforts if the Data Availability section included the range of newly added Genbank accession numbers. These were not apparent in Table S1. (3) I could not find a legend for abbreviations used to identify voucher specimens in Table S1. What does FIELD mean? FMNH or a released specimen? (4) I could not find a table containing elevational ranges of the various taxa included in the analysis, their body sizes, locomotory mode, and geographic range size. their

Experimental design

Remarkably comprehensive, utilizing four tetrapod classes to address the evidence for co-distributed patterns.

Because the authors acknowledge that (a) the orogenic elevation of the EC-MA was asynchronous, and (b) that it varies along its length as a barrier to distribution (in m.a.s.l.) and is punctured by low elevation passes, it would be interesting to know if (1) the test species (or species pairs) occur on either side of these passes, and (2) if there are shared haplotypes between them (and which phylogenies are involved). Shared mtDNA haplotypes for taxa thought to have diverged in the Miocene might reflect more recent or ongoing introgression events

Validity of the findings

The only determination in this paper that gives me pause is the argument that the EC-MA reached effective heights ("significant" as used in the abstract should be replaced) 38-33 Mya. As the authors acknowledge in the Discussion, it is not the highest heights of the EC-MA that are relevant to their test of vicariance--it is the elevation of the passes. When did the passes reach their current elevation? (and why was a 2000 m range maximum chosen if one of the passes is currently 1900 m?)

On line 95, the authors state "Hoorn et al. (2010) shows that Andean uplift was a gradual process that occurred during more than 20 million years..." However, with reference to the Northern Andes, Hoorn et al actually state "However, the most intense peaks of Andean mountain building followed during the late middle Miocene (~12 Ma, Fig. 1C) and early Pliocene (~4.5 Ma, Fig. 1E and figs. S3 to S5)." I am not enough of a geophysicist to challenge their orogenic interpretations, but think they should carefully examine their assertions, particularly with respect to their timing, as they contrast with most traditional interpretations.

Additional comments

When the authors discuss dispersal versus vicariance in the Introduction (e.g. line 45), they might also point out that dispersal models don't generate predictions for other species--they are idiosyncratic. On the other hand, vicariance models yield general area cladograms and are open to tests with co-distributed species.

The legend to Fig 4 states the median divergence time for mammals was 1.03 Ma, yet the dotted line appears at >1.5 Ma...

Reviewer 4 ·

Basic reporting

This study represents a novel approach to integrating geological asynchronous change with lowland tetrapod vicariance in Colombia. The authors test whether the classical model of speciation in the region was due strictly to the northern Andes uplift, rather than having any impact from species’ life history and physiological characteristics on the montane barrier. The integration of geological changes into their models was cohesive with the biological data, and helped to illuminate some of the patterns observed in the models explaining the variance in divergence time.

The geological background of the area was appropriately described in the introduction, and the prose flowed well from the hypotheses into the methods section. The geological history of the area is complex, and needed a thorough description for the reader to understand the variation of geological barriers on the ground. As the authors clearly took great care in their editing of t the manuscript prior to submission, I do not recommend any major structural changes to these sections.

This study provides a valuable increase in understanding for the impact of physiological and life history variation as it interacts with varying times of formation of physical geographic barriers.

Figures:
Figures are all clear with good representation of the geographic and paleogeologic area for the reader. The descriptions of the subsampling were made clearer by Figure 3. Figure 4 clearly denoted estimated time of divergence by the four lowland groups, and Figure 5’s breakdown of potential thermoregulatory effects on divergence time was clearly communicated.

However, Figure 1 has a minor change needed, in that both 75ºW and 70ºW longitude need to be in black rather than in white, and perhaps both the ticks and font size for latitude and longitude could be a bit larger.

Tables:
Tables sufficiently described the sequences used for the 37 species, along with the assumptions of each model, organizing by AIC. 95% CI of each species’ time to divergence was important to include to allow for adequate interpretation of the results of those models.

Supplemental Tables, Figures, and Appendices all appear to be in order.

Line-by-Line Notes:

Line 417: “km2” should be “km2” with the 2 as a superscript or written out as square kilometers (this does not show easily in PeerJ's review system).

Line 577: It seems that utilizing a single reference (Ghalambor, 2006) for thermal physiological responses in lowland vs. tropical poikilotherms is a bit thin here. There has been much work done on the variation in this area, from anurans onward, that can inform this statement. I would suggest that the authors strengthen this connection to their findings with more literature.

Experimental design

The authors carefully constructed both the assumptions of the genetic data included in their model, and the caveats of those data in constructing their estimates. They thoroughly explain their reasoning at each step.

Validity of the findings

The interpretation of the findings appear to be aligned with asynchronous divergence, and the resolution provided here above the usual 5-2 Ma divergence event provided a meaningful insight into divergence in the northern Andes. It will be a valuable start to exploring these hypotheses for other taxonomic lineages not covered here.

Additional comments

I have no additional comments not stated above.

---

## Round 0.2 · accepted · Accept

You have addressed the reviewer comments to an appreciable degree and hence meet the editorial requirements for acceptance. I am delighted to accept your excellent paper.

·

Basic reporting

The authors have done a great job with the feedback received

Experimental design

I have no remaining criticisms

Validity of the findings

PeerJ should publish it!

Additional comments

None

Reviewer 4 ·

Basic reporting

The authors updated their work well in regards to the reviewers’ comments. I want to reiterate that the geological background of the area was appropriately described in the introduction, and very important to the context of this work. Other reviewers thought that it needed to be shortened, but the geological history in that part of the world is complex, and therefore, merits some depth of explanation. Where there is ambiguity in the geological timing, this is now noted by the authors.

This study provides a valuable increase in understanding for the impact of physiological and life history variation as it interacts with varying times of formation of physical geographic barriers.

Figures:
Figures are all clear with good representation of the geographic and paleogeologic area for the reader. They were improved in response to my comments and those of other reviewers.

Tables:
Tables sufficiently described the sequences used for the 37 species, along with the assumptions of each model. No further comment.

Supplemental Tables, Figures, and Appendices all appear to be in order, except that Appendices S3 and S4 still have tracked changes present in the files. The addition of the “newly generated sequence” column as suggested by Reviewer 3 was welcome in Table S1.

Line-by-Line Notes:
None.

Experimental design

I have no additional comments beyond my initial review. The authors thoroughly explain their reasoning at each step.

Validity of the findings

I have no further comments beyond my initial review: The interpretation of the findings appear to be aligned with asynchronous divergence, and the resolution provided here above the usual 5-2 Ma divergence event provided a meaningful insight into divergence in the northern Andes. It will be a valuable start to exploring these hypotheses for other taxonomic lineages not covered here.

Additional comments

I have no additional comments not stated above.The authors took great care in updating and responding to the comments by the reviewers, and the additional details they provide improved the manuscript.